# Calibrated ensembles - a simple way to mitigate ID-OOD accuracy tradeoffs

## Abstract

We often see undesirable tradeoffs in robust machine learning where out-of-distribution (OOD) accuracy is at odds with in-distribution (ID) accuracy. A "robust" classifier obtained via specialized techniques like removing spurious features has better OOD but worse ID accuracy compared to a "standard" classifier trained via vanilla ERM. On six distribution shift datasets, we find that simply ensembling a standard and a robust model is a strong baseline—we match the ID accuracy of a standard model with only a small drop in OOD accuracy compared to the robust model. However, calibrating these models in-domain surprisingly improves the OOD accuracy of the ensemble and eliminates the tradeoff and we achieve the best of both ID and OOD accuracy over the original models.

## 1 Introduction

Machine learning models typically suffer large drops in accuracy in the presence of distribution shift where the test distribution is different from the training distribution. As ML systems are widely deployed, it is important to train models that achieve good "out-of-distribution" (OOD) accuracy. For example, models trained on medical data from a few hospitals should work well when deployed broadly (Zech et al., 2018; AlBadawy et al., 2018). Similarly, when predicting poverty from satellite imagery, models trained on data from a few countries should work well on all countries, particularly those where labels are scarce due to resource constraints (Jean et al., 2016). There has been a lot of research interest in tackling this robustness problem under various settings such as robustness to spurious correlations (Heinze-Deml & Meinshausen, 2017; Sagawa et al., 2020a; Chen et al., 2020b; Liu et al., 2021), robustness to adversarial perturbations (Goodfellow et al., 2015; Madry et al., 2018), domain generalization (Arjovsky et al., 2019; Sun & Saenko, 2016), robustness to demographic shifts (Hashimoto et al., 2018; Duchi et al., 2019) among others. Almost universally across these different settings, an unfortunate tradeoff arises. Robustness interventions typically improve the OOD accuracy but simultaneously cause a drop in the "in-distribution" (ID) accuracy on new test points from the original distribution.

This tradeoff is a major hurdle in using the multitude of proposed robustness interventions. In practice, most inputs are likely to be ID, so it is unsatisfactory to use a "robust" model that has high OOD performance but performs less accurately on these majority ID points. On the other hand, "standard models" (trained without robustness interventions) fail catastrophically in the presence of even small shifts, and it can be highly dangerous to use a standard model even if OOD points are rare. In this work, we ask *is there a general strategy by which we can achieve high accuracy both in-distribution and out-of-distribution and mitigate tradeoffs arising in robustness?*

We consider four benchmark datasets (DomainNet, CIFAR → STL, ImageNet → ImageNet-R, and BREEDS-Entity-30) and two real world satellite remote sensing datasets (Landcover and Cropland), that have been used in prior work on robustness. Our work spans different types of robustness interventions (projecting out spurious correlations, zero-shot language prompting, freezing pretrained features), data modalities (image and time series data), and model architectures (vision transformers, deep convolutional networks, time series convolution). Averaged across these datasets, robustness interventions increase OOD accuracy from 65% to 77%, but decrease ID accuracy from 88% to 85%.

We first explore the natural strategy of ensembling the standard and robust models to combine their strengths. Concretely, we add the probabilities of each model to obtain a prediction with the hope

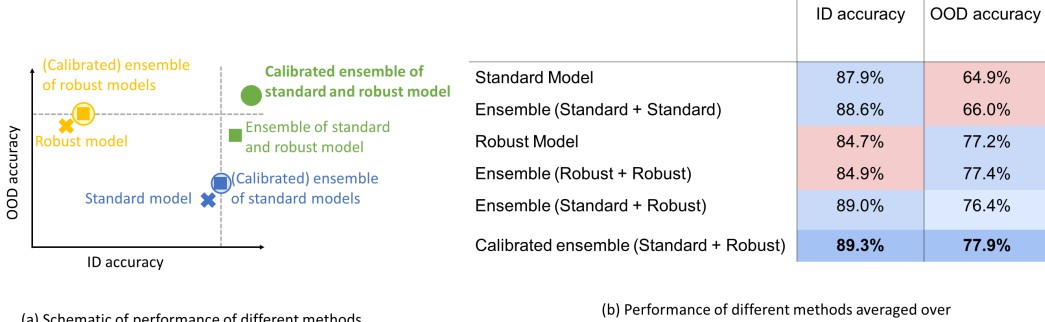

| | ID accuracy | OOD accuracy |
|---|---|---|
| Standard Model | 87.9% | 64.9% |
| Ensemble (Standard + Standard) | 88.6% | 66.0% |
| Robust Model | 84.7% | 77.2% |
| Ensemble (Robust + Robust) | 84.9% | 77.4% |
| Ensemble (Standard + Robust) | 89.0% | 76.4% |
| Calibrated ensemble (Standard + Robust) | **89.3%** | **77.9%** |

(a) Schematic of performance of different methods

(b) Performance of different methods averaged over 6 real distribution shifts

Figure 1: In many settings, we have a 'standard' model that performs better in-distribution, and a 'robust' model that performs better out-of-distribution. Simply ensembling these two models (e.g., by adding their probabilities), gets better ID accuracy than the standard and robust models, and closes most of the OOD gap. Calibrating the models in-distribution (no access to OOD data) before ensembling them leads to further improvements. Note that ensembling two standard or two robust models does not close the gap and only leads to small improvements.

that when the two models conflict, the more confident model (with larger probability) dictates the final prediction. We find that this surprisingly simple baseline already perfoms quite well—on average across all our datasets, this closes $90\%$ of the gap between the OOD of standard models, while outperforming both models ID. In other words, this simple baseline improves the OOD accuracy over standard models without hurting ID accuracy unlike previous robustness interventions. However, vanilla ensembling still leaves a gap as it underperforms the robust model OOD.

We find that simply calibrating both models ID (adjusting their predicted confidence to match their accuracy, on *in-distribution* data) before ensembling them closes this gap. *Calibrated ensembles get an average accuracy of 89.3% ID and 77.9% OOD, and outperform both the standard and robust model, ID and OOD*. The other method in the literature to alleviate robustness induced tradeoffs is self-training that uses large amount of unlabeled data (Raghunathan et al., 2020; Xie et al., 2021; Khani & Liang, 2021). On the two remote-sensing datasets with additional unlabeled data, we find that calibrated ensembles match self-training on these datasets without requiring any unlabeled data. This shows that calibrated ensembles, though conceptually simple, can be highly effective in mitigating tradeoffs. As a sanity check, we find that the method also works when there is no tradeoff: even when the standard model (or robust model) dominates the robust model (or standard model) both ID and OOD, the calibrated ensemble accuracy matches up with the better model in both domains.

While our method is intuitive in a way, it is also intriguing that it works so well because ensembling seems to rely on good uncertainty estimates while it is common wisdom that uncertainty estimates of deep networks are unreliable out-of-distribution (Ovadia et al., 2019). Furthermore, it has been shown that calibration in-domain does not fix the issue of poor uncertainty estimates out-of-distribution. Indeed, on the six datasets we test on, the models fare poorly on standard uncertainty metrics OOD, even after calibrating ID. The expected calibration error (which roughly measures the difference between the model's confidence and accuracy) of the standard model across all datasets is $11\%$. A partial answer might be that even if the models have high calibration error on average, what determines the quality of ensembling is the relative confidence between the standard and robust model on a particular data point. For example, on the remote sensing dataset (Landcover), the standard model is on average $6\%$ more confident in its OOD predictions than the robust model, even though the standard model is less accurate OOD—which seems bad. But at the granularity of individual points, we find the more confident model is more likely to be accurate, enabling calibrated ensembles to achieve higher OOD accuracy than both standard and robust models. While most prior work on confidence estimates has focused on a single model at a population level, our results suggest that examining individual data points and relative confidence of different models is an exciting line of future work.

**Comparison with regular ensembling.**    Ensembling is an extremely common trick in ML to boost performance. However, when using ensembles, it is typical to have each ensemble member obtained by running the same training process with a different random seed which controls the stochastic aspects of the training algorithm like initialization, ordering of the training samples in a batch, which features are selected when using drop out and so on. The idea behind ensembling is that with different random seeds, the algorithm might converge to different solutions with decorrelated errors. In this work, we ensemble a standard and robust model which are obtain by *minimizing entirely different training losses*. This small but subtle change is very crucial. Experimentally, we find that regular ensembling of two standard models does poorly OOD (11% worse than a single robust model), and ensembling two robust models does poorly ID (3% worse than a single standard model). For these regular ensembles, calibration has no effect—the ID and OOD accuracies stay the same.

Why is our proposed ensembling so much better than regular ensembling? Consider the case of spurious correlations as studied in (Sagawa et al., 2020b) where standard classifiers use spurious features irrespective of their random initialization. In other words, all members of the regular ensemble would have the same failure mode. The intuition of decorrelated error does not pan out with regular ensembles when standard models make systematic errors, as is common in most distribution shift settings. On the other hand, in our ensemble, we have a standard model that uses spurious features to be more accurate but also a robust model trained specifically not to use spurious features. By combining these "diverse" models with qualitatively different failure modes, we are able to achieve the best of both ID and OOD performance.

In summary, we show that a simple method of calibrating and then ensembling can combine the best of standard and robust models, getting high ID and OOD accuracy—and calibration plays a key role in this method. This is a general practical method to alleviate tradeoffs which can likely be combined with other innovations to completely eliminate robustness induced tradeoffs in practice. This work also raises several interesting conceptual questions regarding ensembling and uncertainty estimation.

## 2   SETUP

Consider a $K$-class classification task, where the goal is to predict targets $y \in [K]$ from inputs $x \in R^d$.

**Models**: A model $f : \mathbb{R}^d \to \mathbb{R}^k$ takes an input $x \in \mathbb{R}^d$ and outputs $f(x) \in R^k$ where $f(x)_i$ denotes the model's confidence that the output is $y = i$. The model predicts the label $\widehat{y} = \arg\max_k f(x)_k$. The confidences can be converted into probabilities using a softmax function:

$$\text{softmax}(f(x))_i = \frac{\exp(f(x)_i)}{\sum_{j=1}^{k} \exp(f(x)_j)} \qquad \text{softmax} : \mathbb{R}^k \to \mathbb{R}^k \tag{2.1}$$

**Training data**: Let $P_{\text{id}}$ and $P_{\text{ood}}$ denote the underlying distribution of $(x, y)$ pairs in-distribution and out-of-distribution, respectively. We have a validation set $\{(x_i^{\text{val}}, y_i^{\text{val}})\}_{i=1}^{n_{\text{val}}} \sim P_{\text{id}}$ used for early stopping and calibration, a held-out in-distribution test set $\{(x_i^{\text{test}}, y_i^{\text{test}})\}_{i=1}^{n_{\text{test}}} \sim P_{\text{id}}$, and a held-out out-of-distribution test set $\{(x_i^{\text{ood}}, y_i^{\text{ood}})\}_{i=1}^{n_{\text{ood}}} \sim P_{\text{ood}}$.

**Evaluation**: All methods can use the ID validation set for tuning hyperparameters, early stopping, and calibration. The ID and OOD test set are only used for evaluation. Given a model $f$, the ID and OOD accuracies are the average accuracies on these test sets:

$$\text{ID-Acc}(f) = \frac{1}{n_{\text{test}}} \sum_{i=1}^{n_{\text{test}}} \mathbb{I}(y_i^{\text{test}} = \arg\max_k f(x_i^{\text{test}})_k) \tag{2.2}$$

$$\text{OOD-Acc}(f) = \frac{1}{n_{\text{ood}}} \sum_{i=1}^{n_{\text{ood}}} \mathbb{I}(y_i^{\text{ood}} = \arg\max_k f(x_i^{\text{ood}})_k) \tag{2.3}$$

## 3 METHODS

**Calibrated ensembles.** Given a standard models $f_{\text{std}}$ and robust model $f_{\text{rob}}$, calibrated ensembles first calibrate each model using temperature scaling (Guo et al., 2017) with the cross-entropy loss $l$ on the *in-distribution* validation data:

$$T_{\text{std}} = \arg\min_T \frac{1}{n_{\text{val}}} \sum_{i=1}^{n_{\text{val}}} l\Big(\frac{f_{\text{std}}(x_i^{\text{val}})}{T}, y_i^{\text{val}}\Big) \tag{3.1}$$

$$T_{\text{rob}} = \arg\min_T \frac{1}{n_{\text{val}}} \sum_{i=1}^{n_{\text{val}}} l\Big(\frac{f_{\text{rob}}(x_i^{\text{val}})}{T}, y_i^{\text{val}}\Big) \tag{3.2}$$

The goal of temperature scaling is to adjust each model's confidence on the in-distribution validation data. We then ensemble the two models by adding up the probabilities that they predict (Lakshminarayanan et al., 2017). So given an example $x$, we simply output the average $\widehat{p}$ of the probabilities predicted by the standard and robust models after scaling by $T_{\text{std}}$ and $T_{\text{rob}}$ respectively.

$$\widehat{p} = \frac{1}{2}\Big(\text{softmax}\Big(\frac{f_{\text{std}}(x)}{T_{\text{std}}}\Big) + \text{softmax}\Big(\frac{f_{\text{rob}}(x)}{T_{\text{rob}}}\Big)\Big) \tag{3.3}$$

**Other ensembles.** As baselines, we consider a number of additional ensembles.

*Vanilla Ensembling*: Here we simply return the average of the standard and robust model's probabilities, without temperature scaling.

$$\widehat{p} = \frac{1}{2}(\text{softmax}(f_{\text{std}}(x)) + \text{softmax}(f_{\text{rob}}(x))) \tag{3.4}$$

*Tuned Ensembling*: As a potentially stronger baseline, we consider outputting a weighted average of the standard and robust model's probabilities, where the weight $\alpha \in [0, 1]$ is tuned to maximize accuracy on the in-distribution set.

$$\widehat{p} = \alpha\text{softmax}(f_{\text{std}}(x)) + (1 - \alpha)\text{softmax}(f_{\text{rob}}(x))) \tag{3.5}$$

*Alternative combination methods*: We could also combine the logits of the models before taking the softmax, or we could simply output the prediction of the more confident model. We chose to add up the probabilities since that is commonly done by prior work on ensembling (Lakshminarayanan et al., 2017) and we found it to work slightly better than these alternatives.

*Other models*: Each of these ensembling methods takes two models and combines them. As such, we can apply each of these ensembling methods to two standard models or two robust models as well.

## 4 DATASETS

We run experiments spanning three different types of robustness interventions: projecting out spurious metadata, language prompting, and freezing pretrained features. These experiments span multiple data modalities and model architectures.

### 4.1 SPURIOUS METADATA

We run experiments on two remote sensing datasets used in prior work studying ID-OOD trade-offs (Xie et al., 2021). These datasets consist of a core input $x$ (image data or time series data) and metadata $z$ (e.g., location, meteorological climate data). The metadata is spuriously correlated with the target—using the metadata to predict labels improves accuracy in-distribution (ID), but hurts accuracy out-of-distribution. Xie et al. (2021) consider a standard model that takes in both the core inputs and metadata to predict the target, and a robust model that only takes in the core inputs and does some additional pretraining. They call these the 'aux-in' and 'aux-out' models respectively.

**Cropland.** The goal is to predict whether a satellite image is of a cropland or not. The core input $x$ is an RGB satellite image, and the metadata $z$ consists of location coordinates and vegetation bands. The original dataset is from Wang et al. (2020), and we use U-net model checkpoints from Xie et al. (2021).

**Landcover.** The goal is to predict the land type from satellite data at a given location. Here, the core input $x$ is a time series measured by NASA's MODIS satellite (Vermote, 2015), and $z$ is climate data (e.g., temperature) at that location. The dataset is from Gislason et al. (2006); Rußwurm et al. (2020). We use model checkpoints from Xie et al. (2021) where they use 1D convolutions for time series data.

## 4.2 ZERO-SHOT LANGUAGE PROMPTING

Radford et al. (2021) (CLIP) pretrain a model on a large multi-modal language and vision dataset. The model can then predict the label of an image by comparing the image embedding, with the language embedding for prompts such as 'photo of an apple' or 'photo of a banana'. They show that this zero-shot language prompting approach can be much more accurate out-of-distribution than the traditional method of fine-tuning the entire model.

**ImageNet → ImageNet-R.** We use a CLIP vision transformer, specifically a ViT-B/16, which is the best publicly available model. The robust model uses language prompts to make zero-shot predictions on ImageNet-Renditions (Hendrycks et al., 2020), a dataset containing cartoon, graffiti, video game, etc, renditions of ImageNet classes. The standard model initializes with weights from the CLIP model, and fine-tunes on ImageNet (Russakovsky et al., 2015) training data for 10 epochs with a batch size of 64, initial learning rate of 0.0001 with a cosine learning rate decay, before making predictions on ImageNet-R. We note that the robust model gets 10% lower accuracy ID (on ImageNet validation examples), but gets 30% higher accuracy OOD (on ImageNet-R test examples)

## 4.3 FREEZING PRETRAINED FEATURES

When adapting a pretrained model to an ID dataset, typically all the model parameters are fine-tuned. Recent work looks at 'lightweight' fine-tuning, where only parts of the model are adapted—this can often do better OOD even though the ID performance is worse (Li & Liang, 2021; Houlsby et al., 2019). We consider three distribution shift datasets where the standard model starts from a pretrained initialization and fine-tunes all parameters on an ID dataset, and the robust model only learns the top linear 'head' layer.

**DomainNet.** A standard domain adaptation dataset (Peng et al., 2019). Here, our ID dataset contains 'sketch' images (e.g., drawings of apples, elephants, etc), and the OOD dataset contains 'real' photos of the same categories. We use the version of the dataset from Tan et al. (2020). We start from a CLIP pretrained ResNet50 and either fine-tune for 50 epochs with batch size 64 and learning rate 0.001 with cosine learning rate decay (to get a standard model) or train the head layer using sklearn logistic regression (to get a robust model).

**CIFAR-10 → STL.** Another standard domain adaptation dataset (French et al., 2018), where the ID is CIFAR-10 (Krizhevsky, 2009), and the OOD is STL (Coates et al., 2011). We start from a ResNet50 pretrained on unlabeled ImageNet examples using MoCo-v2 (Chen et al., 2020a) and either fine-tune for 20 epochs with a batch size of 64 and learning rate of 0.001 with cosine learning rate decay (to get a standard model) or train the head layer using sklearn logistic regression (to get a robust model).

**Living-17.** Part of the BREEDS benchmark (Santurkar et al., 2020), here the goal is to classify an image as one of 17 animal categories such as 'bear'—the ID dataset contains images of black bears and sloth bears and the OOD dataset has images of brown bears and polar bears. We start from a ResNet50 pretrained on unlabeled ImageNet examples using MoCo-v2 (Chen et al., 2020a) and either fine-tune for 20 epochs with batch size 64 and learning rate of 0.001 with cosine learning rate decay (to get a standard model) or train the head layer using sklearn logistic regression (to get a robust model).

Table 1: *Out-of-distribution (OOD)* accuracies for the standard model, robust model, and calibrated ensembles, across six datasets. Calibrated ensembling matches or outperforms the better model in 5/6 cases, and on average outperforms both the standard and robust models. For the remaining dataset, DomainNet, calibrated ensembles close 96% of the gap between the standard and robust model.

|  | Ent30 | DomNet | STL | Land | Crop | ImNet-R |
|---|---|---|---|---|---|---|
| Standard | 60.7 (0.1) | 55.3 (0.4) | 82.4 (0.3) | 55.7 (1.1) | **85.6 (5.8)** | 49.6 (-) |
| Robust | 63.2 (1.1) | **87.2 (1.1)** | 85.1 (0.2) | **60.4 (1.1)** | 89.8 (0.4) | **77.5 (-)** |
| Cal ensemble | **64.7 (0.5)** | 86.1 (0.2) | **87.3 (0.2)** | **60.8 (0.8)** | 91.3 (0.8) | 77.1 (-) |

Table 2: *In-distribution (ID)* accuracies for the standard model, robust model, and calibrated ensembling, across six datasets. Calibrated ensembling matches or outperforms the better model in 5/6 cases, and on average outperforms both the standard and robust models. For the remaining dataset, CIFAR-10, calibrated ensembles close 97% of the gap between the standard and robust model.

|  | Ent30 | DomNet | CIFAR10 | Land | Crop | ImNet |
|---|---|---|---|---|---|---|
| Standard | **93.6 (0.2)** | 83.9 (1.0) | **97.4 (0.1)** | **76.9 (0.3)** | 95.3 (0.0) | **80.5 (-)** |
| Robust | 90.7 (0.2) | 89.2 (0.1) | 92.0 (0.0) | 72.7 (0.2) | 95.1 (0.1) | 68.4 (-) |
| Cal ensemble | **93.7 (0.1)** | **91.2 (0.7)** | 97.2 (0.1) | **77.2 (0.2)** | **95.6 (0.1)** | **81.1 (-)** |

## 5 EXPERIMENTS

### 5.1 MAIN RESULTS

**Strong ID and OOD accuracy**: Calibrating and then ensembling a standard and a robust model, gets the best of both worlds, typically outperforming the standard and robust model both ID (Table 2) and OOD (Table 1). Averaged across the datasets, calibrated ensembles get 89.3% ID (vs 87.9% for the standard model and 84.7% for the robust model) and 77.9% OOD (vs 77.2% for the robust model and 64.9% for the standard model). The method works across the board—calibrated ensembles achieve the best performance on 5/6 ID datasets, and on 5/6 OOD datasets. For the remaining two datasets, DomainNet OOD and CIFAR-10 ID, calibrated ensembles close over 95% of the gap between the standard and robust model (96% for DomainNet, 97% for CIFAR-10).

**Competitive with self-training**: The remote sensing datasets have lots of unlabeled data so prior work uses self-training on these datasets to mitigate the ID-OOD accuracy tradeoff—we take check-points from the official implementation in (Xie et al., 2021) and compare calibrated ensembles with self-training. Table 3 shows that calibrated-ensembles match or outperform self-training on both datasets, both ID and OOD. We believe this is an interesting result because calibrated ensembling is a simple method and does not need additional unlabeled data.

### 5.2 ENSEMBLING ABLATIONS

We run ablations to show the importance of calibration, and of including both a robust and a standard model in the ensemble. We find that a strong baseline of tuning the ensemble weights on ID data has lower accuracy than calibrated ensembles OOD.

**Standard ensembles do not mitigate tradeoffs**: Ensembling two standard models or two robust models does not mitigate the ID-OOD accuracy tradeoff. As seen in Figure 1, ensembling two standard models gets an average OOD accuracy of 66.0%, which is substantially worse than our method (77.9%) or even a single robust model (77.2%). Calibrating the standard models does not change its accuracy. Similarly calibrating and ensembling two robust models gets an average ID accuracy of 84.9% which is worse than our method (89.3%) or even a single standard model (87.9%). In summary, calibrated ensembles outperform standard and robust models, ID and OOD (Figure 1).

Table 3: Calibrated ensembles are competitive with self-training (Xie et al., 2021) ID and OOD, which requires unlabeled data.

|  | Cropland | | Landcover | |
| --- | --- | --- | --- | --- |
|  | ID Acc | OOD Acc | ID Acc | OOD Acc |
| Standard model | 95.3 (0.0) | **85.6 (5.8)** | 76.9 (0.3) | 55.7 (1.1) |
| Robust model | 95.1 (0.1) | 89.8 (0.4) | 72.7 (0.2) | **60.4 (1.1)** |
| Self-training | 95.3 (0.2) | **90.6 (0.6)** | **77.0 (0.4)** | **61.0 (0.7)** |
| Cal ensembling | **95.6 (0.1)** | **91.3 (0.8)** | **77.2 (0.2)** | **60.8 (0.8)** |

Table 4: *OOD* accuracies: calibrated ensembles outperform vanilla ensembles and even tuned ensembles where the combination weights are tuned to maximize in-distribution accuracy. Averaged across the datasets, calibrated ensembles get an OOD accuracy of 74.6%, while tuned ensembles get an accuracy of 71.3%. The in-distribution accuracies of the methods are very close (within 0.2% of each other).

|  | Ent30 | DomNet | STL | Land | Crop | ImNet-R |
| --- | --- | --- | --- | --- | --- | --- |
| Vanilla Logits | **64.9 (0.3)** | 75.7 (1.2) | **87.3 (0.2)** | **60.5 (0.8)** | **90.9 (0.2)** | 72.2 (-) |
| Vanilla Probs | **64.6 (0.4)** | 78.7 (1.3) | 87.2 (0.2) | 59.5 (1.0) | **90.9 (0.2)** | **77.4 (-)** |
| Tuned Logits | **64.6 (0.6)** | 86.3 (0.6) | 85.7 (0.9) | 58.7 (1.2) | 87.3 (5.7) | 63.1 (-) |
| Tuned Probs | 62.8 (0.7) | **86.9 (0.2)** | 85.0 (1.3) | 58.7 (2.2) | 86.8 (5.5) | 63.8 (-) |
| Calibrated Logits | **65.0 (0.4)** | 84.4 (0.3) | **87.5 (0.2)** | **61.2 (0.8)** | **91.3 (0.8)** | 71.7 (-) |
| Calibrated Probs | **64.7 (0.5)** | 86.1 (0.2) | **87.3 (0.2)** | **60.8 (0.8)** | **91.3 (0.8)** | 77.1 (-) |

**Calibration is important**: We compared calibrated ensembles with vanilla ensembles (no calibration) and tuned ensembles where how to weight the predictions of the standard versus robust model is learned on ID data. Interestingly, calibrated ensembles substantially outperform tuned ensembles OOD (calibrated ensembles: 77.9%, tuned ensembles: 74.0%) with less than a 0.1% drop in ID accuracy relative to tuned ensembles. Naturally, we expect the tuned ensemble to do the best ID since its weights are tailored for ID—what is surprising is that the calibrated ensembles do so much better OOD without using any OOD data either. Calibrated ensembles outperform vanilla ensembles both ID and OOD as well. We show the accuracies for the different types of ensembles in Table 4 (OOD) and Table 5 (ID)—we also ablate between combining the probabilities versus the logits (before softmax) of the standard and robust models.

## 5.3 CALIBRATION AND RELATIVE CALIBRATION

In this section we examine the calibration and relative calibration of our models, which provides a partial intuitive explanation for the success of calibrated ensembles. Since we calibrated the standard and robust models on in-distribution (ID) data, we expect them to be calibrated even on held out ID test data—this follows from standard statistical guarantees for calibration. Table 7 confirms this intuition—the ECE of the standard and robust model are very low ID, on average the ID ECE is 1.2% for the standard model and 2.0% for the robust model. However, prior work shows that the calibration of models degrades substantially with distribution shift—indeed, Table 6 shows that the OOD ECE of the standard (average: 11.0%) and robust (average: 7.3%) models are much higher.

Why do calibrated ensembles mitigate the ID-OOD tradeoff even though neural networks are not calibrated OOD? We show that *relative calibration* provides a partial answer towards this. The final prediction is typically made by the model that is more confident. So if the robust model is more confident than the standard model OOD then the ensemble inherits the high OOD accuracy of the robust model—even if the absolute ECE of the standard and robust models are both bad.

To get a handle at relative calibration we measure 1. the *accuracy gap* between the standard and robust model, which is the robust model's accuracy minus the standard model's accuracy, and 2. the *confidence gap* between the two models: the robust model's average confidence minus the standard

Table 5: *ID* accuracies: The in-distribution accuracies of calibrated ensembles, tuned ensembles, and vanilla ensembles are very close (within confidence intervals), so any of these methods are acceptable if we are looking at in-distribution accuracy. However, they perform quite differently when it comes to OOD accuracy (Table 4).

|  | Ent30 | DomNet | CIFAR10 | Land | Crop | ImNet |
|---|---|---|---|---|---|---|
| Logits | 93.7 (0.1) | 89.3 (0.6) | 97.3 (0.1) | 77.4 (0.1) | 95.5 (0.1) | 80.9 (-) |
| Probs | 93.7 (0.1) | 89.1 (0.4) | 97.3 (0.1) | 77.4 (0.2) | 95.5 (0.1) | 81.0 (-) |
| Tuned Logits | 93.8 (0.0) | 91.3 (0.2) | 97.4 (0.1) | 77.3 (0.4) | 95.6 (0.1) | 81.7 (-) |
| Tuned Probs | 93.8 (0.1) | 90.6 (0.7) | 97.4 (0.1) | 77.1 (0.3) | 95.5 (0.1) | 81.3 (-) |
| Calibrated Logits | 93.7 (0.1) | 91.1 (0.4) | 97.2 (0.1) | 77.2 (0.2) | 95.6 (0.1) | 81.0 (-) |
| Calibrated Probs | 93.7 (0.1) | 91.2 (0.7) | 97.2 (0.1) | 77.2 (0.2) | 95.6 (0.1) | 81.1 (-) |

Table 6: *OOD* ECE: The expected calibration error (ECE) of the standard and robust models on OOD test data, after post-calibration in ID validation data. The calibration errors here are high, especially compared to the ID calibration errors in Table 7.

|  | Ent30 | DomNet | STL | Land | Crop | ImNet-R |
|---|---|---|---|---|---|---|
| Calibrated Standard | 15.4 (0.8) | 13.6 (1.5) | 5.6 (1.1) | 16.4 (0.8) | 7.4 (4.8) | 7.8 (-) |
| Calibrated Robust | 14.3 (1.5) | 5.5 (0.5) | 8.2 (0.0) | 6.5 (1.1) | 5.0 (0.3) | 4.0 (-) |

model's average confidence. Ideally, the two should align with each other and at least have the same sign: if the robust model is more accurate then it should also be more confident. In Table 8 we show that this provides a reasonable initial explanation for the success of calibrated ensembles. The confidence gap and accuracy gap tend to have the same sign, and calibration aligns the two better. However, calibrated ensembles still work on LandCover—even though the relative calibration average across the entire dataset is not correct (the standard model is more confident but less accurate OOD) at the granularity of individual points ensembling is able to get the benefits of both the standard and robust models.

Why might calibrating models ID improve the relative calibration OOD? Standard deep learning models tend to be overparameterized, and can often get near 0 loss on the training data. As such they tend to be highly overconfident on ID and OOD data. Robustness interventions typically involve additional constraints (projecting out spurious input features, lightweight fine-tuning, extensive data augmentation) and so robust models generally tend to be less overconfident. Calibrating both models, even just ID, can make their confidences more comparable—even though both models degrade OOD, the hope is that the relative calibration continues to track their accuracy.

We note that all this is intuition (theory about OOD calibration and ensembling is scarce and highly challenging)—we hope that future work formalizes these arguments. We think relative calibration might be a promising future direction—even if models are not calibrated OOD, their relative calibration could be much better.

## 6 RELATED WORKS AND DISCUSSION

**Calibration.** Calibration has been widely studied in machine learning (Naeini et al., 2014; Guo et al., 2017; Kumar et al., 2019), and applications such as meteorology (Murphy, 1973; DeGroot & Fienberg, 1983; Gneiting & Raftery, 2005), fairness (Hebert-Johnson et al., 2018), and healthcare (Jiang et al., 2012). Many of these works focus on the in-distribution (ID) setting, where models are calibrated on the same distribution that they are evaluated on. Ovadia et al. (2019) show that if we calibrate (e.g., via temperature scaling) a model ID, it still has poor uncertainties OOD. Jones et al. (2021) also show that model uncertainties can be quite unreliable out-of-distribution. However, we show that despite having poor uncertainties on traditional metrics, calibrated models can be combined effectively to mitigate ID-OOD tradeoffs.

Table 7: *ID* ECE: The expected calibration error (ECE) of the standard and robust models on ID test data, after post-calibration in ID validation data. The calibration errors are fairly low—note that we only use 500 examples to temperature scale, so for ImageNet we have fewer examples than classes for post-calibration, but the models are still fairly well calibrated.

|  | Ent30 | DomNet | CIFAR10 | Land | Crop | ImNet |
|---|---|---|---|---|---|---|
| Cal Standard | 0.7 (0.1) | 2.0 (0.3) | 0.8 (0.2) | 1.1 (0.5) | 1.4 (0.3) | 1.0 (-) |
| Cal Robust | 1.1 (0.4) | 2.2 (0.2) | 1.3 (0.2) | 1.7 (0.3) | 3.5 (0.2) | 2.3 (-) |

Table 8: We show the accuracy gap (difference between the accuracy of the standard and robust model) and the confidence gap (difference in confidence between the two) before and after calibration. The accuracy gap and confidence gap typically have the same sign, and this improves after calibrating ID.

|  | Ent30 | DomNet | STL | Land | Crop | ImNet-R |
|---|---|---|---|---|---|---|
| Acc Gap | 2.5 | 31.9 | 2.7 | 4.7 | 4.2 | 27.9 |
| Conf Gap | -2.3 | 3.6 | 1.1 | -13.5 | 4.3 | 18.6 |
| (+ Cal) Conf Gap | 1.4 | 23.7 | 5.5 | -6 | 1.3 | 16.6 |

**Ensembling.** Ensembling models is a common way to get an accuracy boost—typically the ensemble members are trained on the same data, but with a different random seed (Lakshminarayanan et al., 2017) or augmentation (Stickland & Murray, 2020). In the setting where the ensemble members mostly differ by random seeds or augmentations, prior work has shown that calibrating the members of an ensemble does not help (Wu & Gales, 2021; Ovadia et al., 2019). Indeed, we find that calibration has minimal effect when we ensemble two standard, or two robust models, that are trained from different seeds. However when we combine two very different models (standard and robust), calibration leads to clear improvements.

**Mitigating ID-OOD tradeoffs.** Tradeoffs between ID and OOD accuracy are widely studied and prior work self-trains on large amounts of unlabeled data to mitigate such tradeoffs (Raghunathan et al., 2020; Xie et al., 2021; Khani & Liang, 2021). In contrast, our approach uses no extra unlabeled data and is a simple method where we just add up the model probabilities after a quick calibration step. In concurrent and independent work, (Wortsman et al., 2021) show on ImageNet and variants (e.g. ImageNet-v2, ImageNet-sketch, ImageNet-R) that there *exists* a way to ensemble a CLIP zero-shot and fine-tuned model to get good ID and OOD accuracy—however learning exactly how to ensemble the models might require OOD data, which is not available. We show that the natural way to learn how to weight ensemble members—selecting the weights to optimize in-distribution accuracy—does not mitigate the ID-OOD gap, but calibrated ensembles do.

**Discussion.** In this paper, we show that a simple method of calibrating and then ensembling models can eliminate the tradeoff between in-distribution (ID) and out-of-distribution (OOD) accuracy. We hope that this leads to more widespread use and deployment of robustness interventions. One mystery is why calibrated ensembles work so well—especially when tuning ensemble weights ID does not, and the worse standard model can still be more confident OOD.

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
