# OpenReview forum: "Calibrated ensembles - a simple way to mitigate ID-OOD accuracy tradeoffs"
_ICLR.cc/2022/Conference — ICLR 2022 Submitted_

### Official Review · Reviewer_7Ez2 · 2021-10-24

**Correctness:** 3
**Technical Novelty And Significance:** 3
**Empirical Novelty And Significance:** 3
**Recommendation:** 6
**Confidence:** 4

**Main Review:**

This paper has many strengths:
- The idea is impactful, easy to implement, and explained clearly.

However, there are some areas where the paper could be improved:
1) More complete baselines.

A number of baselines including vanilla ensembling, tuned ensembling, and alternative combination methods are mentioned in Section 3. However, their relative ID and OOD performances are not sufficiently explored. Table 4 contains the OOD accuracies for some of these methods for all datasets, but in terms of ID performance only the averaged version is available (Figure 1, right), and the Tuned ID baseline remains absent. There is room to include this as the paper is currently just over seven pages, and it would improve the paper. In general, it would be nice to create Figure 1 (left) using real data for each dataset, as currently it is only shown with fake data I believe.

There are also some baselines that are discussed but absent, e.g., in Section 3 the authors mention that they found combining the softmax probabilities to work better than combining the logits. However, this alternative is absent in all tables / plots. This baseline in particular is interesting for the following reason: I believe that without the softmax, there is a mixing coefficient alpha (e.g., in Eq. 3.5) which exactly matches the proposed method (Eq. 3.3). Going back to the idea from the previous paragraph of creating a scatter plot with real data, it would even be possible to plot the curve from varying alpha and see where the proposed method lies on the curve -- how close is it to the optimal alpha? Calibration could then be interpreted as a way to choose this mixing coefficient, which would be very interesting.

When comparing with vanilla ensembling, this paper use a default mixing coefficient of 0.5. However, this paper mentions that [1] only shows there exists a way to ensemble a standard and robust model, and requires OOD data to learn how to do so. As 0.5 is assumed to be the default mixing coefficient for vanilla ensembling, it could be nice to also compare with [1] using, e.g., mixing coefficient 0.5. This seems relatively straightforward for ImageNet-R as the experimental set-up is very similar. Going back to the previous paragraph, it would be a very interesting comparison if you could find the mixing coefficient alpha via calibration then apply it to the method of [1]. As I understand, the concurrent method of [1] is very similar to this work as both propose to ensemble a robust model with a model that performs good ID and find the benefits of both in the ensemble.

2) Clarity in experimental details.

Clarity in experimental details could strengthen the paper, for instance when fine-tuning on ImageNet for the ImageNet-R experiments which optimizer and batch size are used. Lack of these details will make reproduction difficult. One drawback of this method is that it is more expensive than a single model, but this experimental detail is absent. Moreover, why is the CLIP zero-shot performance on ImageNet-R extremely low? This is substantially lower than what is reported in [1,2] for the same model type (ViT-B/16).

[1] https://arxiv.org/abs/2109.01903
[2] https://arxiv.org/abs/2109.01134

**Summary Of The Paper:**

A robust model attains high accuracy OOD while a standard model attains high accuracy ID. This paper proposes ensembling a robust and standard model to attain high accuracy ID and OOD. In particular, this paper recommends calibrating both the standard and robust models on ID data before ensembling.

**Summary Of The Review:**

This paper explores an interesting idea of ensembling a robust and standard model to attain high accuracy ID and OOD. However, the paper would substantially benefit from a more complete exploration of baselines and thorough explanation of experimental details. For instance, even some of the baselines mentioned are partially included or absent, and accuracy of ImageNet-R is substantially lower than usually reported. I recommend that the paper could substantially benefit from additional exploration, and may have been rushed in current form.

Edit: many of my concerns were addressed and I have changed my score to 6.

---

> ### Author Response · Authors · 2021-11-23
> **Added ablations, comparisons, ImageNet-R clarification**
>
> We thank reviewer EkrF for the detailed review, and for saying that the idea is impactful and has many strengths. We have added additional results and ablations requested by the reviewer, and answer their questions here.
>
> > A number of baselines including vanilla ensembling, tuned ensembling, and alternative combination methods are mentioned in Section 3.
>
> > in terms of ID performance only the averaged version is available (Figure 1, right), and the Tuned ID baseline remains absent.
>
> We apologize for omitting these. We have added in-distribution per-dataset accuracies for vanilla ensembling, tuned ID, and calibrated ensembles in Table 5.
>
> > the authors mention that they found combining the softmax probabilities to work better than combining the logits
>
> > this alternative is absent in all tables / plots
>
> We agree that this ablation is interesting. We initially reported results only for combining the softmax probabilities, since that is what is done in the pioneering work on deep ensembles (see page 5 of https://arxiv.org/pdf/1612.01474.pdf right below the Algorithm 1 box), and appears to be fairly standard.
>
> We have added per-dataset results for combining the logits instead of softmax probabilities in Table 4 (OOD) and Table 5 (ID). We report this for vanilla ensembles, tuned ID, and calibrated ensembles. On average, combining the softmax probabilities works better that combining logits both before and after calibration. For example, for average OOD accuracies:
>
> |                        | Average OOD Accuracy |
> |------------------------|----------------------|
> | Logits                 | 75.3                 |
> | Probs                  | 76.4                 |
> | Logits (+ calibration) | 76.9                 |
> | Probs (+ calibration)  | 77.9                 |
>
> We show results for each dataset in the paper.
>
> >  this paper mentions that [1] only shows there exists a way to ensemble a standard and robust model
>
> > it could be nice to also compare with [1] using, e.g., mixing coefficient 0.5
>
> On page 8 of [1] (https://arxiv.org/pdf/2109.01903.pdf) they discuss "ensembling softmax outputs" and say "This method performs comparably to weight-space ensembling". Ensembling softmax outputs with a weight of 0.5 corresponds to vanilla ensembling in our paper, so we hope this is a reasonable comparison with the concurrent work.
>
> > Clarity in experimental details could strengthen the paper
> > optimizer and batch size are used
>
> We apologize for omitting these, and have added details. If accepted we will release all checkpoints and code with the camera ready.
>
> > why is the CLIP zero-shot performance on ImageNet-R extremely low
>
> This is a great question. In short, there are two different ways of measuring OOD accuracy on ImageNet-R because ImageNet has 1000 classes but ImageNet-R only has a subset of 200 classes. The model f takes in an example x, and outputs a vector \in R^1000, where $f(x)_i$ is the model's confidence that the label is i.
>
> Option 1: $\hat{y} = \mbox{argmax}_i f(x)_i$
>
> Option 2: $\hat{y} = \mbox{argmax}_i f(x)_i$ where i is a valid ImageNet-R label (one of the valid 200 classes in ImageNet-R)
>
> We initially used option 1, whereas the other referenced works use option 2. Option 2 makes sense if we know what classes the OOD data is constrained to. Since option 2 appears to be more standard, in our updated submission we have switched to option 2 and reported all numbers in this regime. For example, under option 2 the zero-shot accuracy of the ViT-B/16 on ImageNet-R is 77.5%

---

> > ### Comment · Reviewer_7Ez2 · 2021-11-24
> > **Thank you.**
> >
> > Thank you for addressing many of my comments, I've raised my score to 6.

---

### Official Review · Reviewer_EkrF · 2021-10-28

**Correctness:** 2
**Technical Novelty And Significance:** 2
**Empirical Novelty And Significance:** 3
**Recommendation:** 5
**Confidence:** 4

**Main Review:**

This is an interesting paper that discusses an important and timely topic, the trade-off between in-distribution and out-of-distribution performance. The paper empirically analyzes this trade-off and shows that standard ERM-based learners perform better ID at the expense of OOD, while robust learners are the opposite. The authors then approach this problem head-on, and propose a simple yet effective solution to this perceived trade-off. The proposed solution relies on resembling both types of models. Then, to further improve performance, the authors calibrate the models and demonstrate substantial improvements.

While the strong empirical results and simple approach are definitely encouraging, I find significant problems with the paper at its current state.

First, the lack of any theoretical contribution or even a substantive intuition into why this approach should work, and when it shouldn’t is highly problematic. This is even more problematic considering existing papers that discuss just that [1]. There seems to be substantial agreement between [1] and this paper, along with some disagreement regarding the connection between ID calibration and OOD performance. Discussing this in the paper and showing how the two might agree will surely improve the paper.

Second, the methods used in the paper can probably be improved. While good performance with a simple method is always a good thing, it would be beneficial to try and further improve performance with a more advanced re-calibration approach or an end-to-end calibrated model. The same critique also stands for the ensemble approaches, which at the moment only consider a combination of two models.

Lastly, the paper is quite thin in analysis and discussion. While I don’t have a particular problem with the lack of novel approaches, I do find it redundant adding an equation for softmax and accuracy. If deleted, the authors should use the 1+ pages remaining space to add a detailed analysis of when and why this approach works in reality. Other potential uses of this space include a representative theoretical analysis, more baselines and experiments on widely used OOD benchmarks such as WILDS [2].


[1] On Calibration and Out-of-domain Generalization
[2] WILDS: A Benchmark of in-the-Wild Distribution Shifts


**Summary Of The Paper:**

This paper discusses the trade-off between in-distribution and out-of-distribution accuracy, where ERM-based learners have good ID but poor OOD performance, and robust learners the opposite. The authors demonstrate that by ensembling both types of learners, and calibrating their ID accuracy, they can outperform the standard and robust models both on ID and OOD data.

The main contribution of the paper is the strong empirical results provided by the simple yet effective approach.


**Summary Of The Review:**

While I do think that this paper discusses an important problem and presents methods that perform reasonably well on real data, I think that its current form the paper is a bit too thin and requires some more work before it is published. My main concerns are the lack of theoretical analysis, as well as additional experiments and analysis with more advanced methods. I also point to a previous publication that might help the authors relate to in their analysis.

---

> ### Author Response · Authors · 2021-11-23
> **Responses on intuition, and other re-calibration methods**
>
> We thank reviewer EkrF for the useful suggestions, and for saying that the paper is "interesting", tackles "an important and timely problem", and shows "strong empirical results" with a "simple approach".
>
> > The reviewer's main concern appears to be "the lack of substantive intuition into why this approach should work"
>
> We believe the observation that this method works is already interesting. First, this method is much simpler but still competitive with existing tradeoff mitigating methods that use unlabeled data. In general, most observations in deep learning are first reported empirically, and it takes many years and papers to build a partial understanding of why these methods work. For example, at present we only have a partial understanding of why deep ensembles work (especially when ensemble members are trained on the same dataset, as opposed to bootstrap resamples), 5-6 years after initial empirical success was reported.
>
> That said, your point is well taken. We have added intuitions and analysis for why the method works in Section 5.3. One key point is that we do not need the models to be calibrated OOD---**a weaker notion of relative calibration** suffices. For example, if the robust model is more accurate and more confident OOD, then we expect that the ensemble will typically use the robust model's predictions OOD. Indeed, Table 8 shows that the relative calibration OOD is much better than the absolute calibration OOD. (ID both the absolute and relative calibration are good)
>
> Why does calibrating ID improve the relative calibration? Standard deep learning models tend to be overparameterized, and can often get near 0 loss on the training data. As such they tend to be highly overconfident on ID and OOD data. Robustness interventions typically involve additional constraints (projecting out spurious input features, lightweight fine-tuning, extensive data augmentation) and so robust models generally tend to be less overconfident. Calibrating both models, even just ID, makes their confidences more comparable. This is particularly important for OOD---for in-distribution data the standard model does better anyway, so its overconfidence is not an issue.
>
> This is only a first step towards understanding this phenomena, but we hope this discussion strengthens the paper.
>
> > The reviewer also asked how this work compares with [1] On Calibration and Out-of-domain Generalization
>
> Our understanding of [1] is: roughly, they show that if you're calibrated on many domains (domains > no. of features) in a linear model, then your features are calibrated (and invariant) on new domains. This is a very interesting piece of work, although in our case **we only have one domain at training time**, which makes the setting quite different.
>
> > The reviewer asked what would happen if we try "a more advanced re-calibration approach or an end-to-end calibrated model" to "further improve performance"
>
> This is a great question. We have a few responses to this:
>
> - **The main bottleneck is OOD calibration error**: We note that after temperature scaling the ID ECE is on average 1-2%, while the OOD ECE is an order of magnitude higher (around 10%). So even if we use a more sophisticated method to shave off 1% from the ID calibration error, we don't expect the OOD calibration to change substantially.
> - This simple method solves the problem and gets the "best of both worlds", the strong ID performance of the standard model and the strong OOD performance of robust models.
>
> > use the 1+ pages remaining space
>
> We were trying to make the paper more concise and easy to read since it is a simple but effective idea. However, we agree with your point and have added a page to dig into why the method works. Thank you for the suggestion.

---

> > ### Comment · Reviewer_EkrF · 2021-11-25
> > **Post-rebuttal Comment**
> >
> > I want to thank the authors for their response to my comments.
> > However, I remain unconvinced of the theoretical support for the arguments made in the paper, and still believe that this paper will benefit from additional experiments and analysis with more advanced methods.

---

> > > ### Author Response · Authors · 2021-11-29
> > > **Added vector scaling (similar results), existing robustness methods not effective on WILDS**
> > >
> > > We thank the reviewer for engaging with us to improve our work. We've looked into more advanced calibration methods, and alternative datasets such as WILDS.
> > >
> > > > The reviewer suggested we examine more powerful calibration methods
> > >
> > > **We've now also run experiments with vector scaling**. The key point is that temperature scaling is particularly amenable to cases where the label space is large (e.g. our ImageNet experiment has 1000 classes but 500 recalibration examples), and more advanced methods don't affect OOD calibration much even when they improve ID calibration.
> > >
> > > - **More advanced calibration methods often require more samples** because they keep at least one parameter per class, and so require at least one example per class. They don't work well on our ImageNet experiment---our recalibration set has 500 examples, while ImageNet has 1000 classes. Vector scaling keeps one parameter per output class, so the values for all the unseen classes are undefined. This is also **true for calibration methods such as vector scaling, matrix scaling, Dirichlet calibration, platt-binning for each class, histogram binning** [1, 2, 3, 4, 5]
> > >
> > > - **While they can improve ID calibration, there's minimal impact on OOD calibration**. Vector scaling does work for our other five experiments. The reviewer's intuition was correct---vector scaling achieves better ID calibration. However, the OOD calibration of vector scaling is actually a bit worse than temperature scaling. **Temperature scaling does slightly better in final accuracy both ID and OOD**.
> > >
> > > We will add this to our manuscript---thank you for the suggestion.
> > >
> > > > The reviewer suggested adding experiments on more datasets, e.g. WILDS.
> > >
> > > This is a good suggestion. The challenge is that right now **most robustness methods simply don't work on WILDS**---the original WILDS paper reports that the robustness methods they tried "generally fail to improve over models trained with ERM". Since WILDS is a new dataset, we hope that new robustness methods will emerge that can be used in future work
> > >
> > > We want to emphasize that while our paper is concise, our experiments are broad. We consider 6 datasets, including popular domain adaptation benchmarks (DomainNet, CIFAR $\to$ STL), robustness benchmarks (BREEDS, ImageNet $\to$ ImageNet-R), and two real world satellite remote sensing datasets including a time series dataset. We consider three robustness interventions, and multiple architectures (vision transformers, ResNet-50, time series convolutions).
> > >
> > > [1] Obtaining calibrated probability estimates from decision trees and naive bayesian classifiers. B. Zadrozny and C. Elkan. ICML 2001.
> > >
> > > [2] On Calibration of Modern Neural Networks. C. Guo, G. Pleiss, Y. Sun, K. Weinberger. ICML 2017.
> > >
> > > [3] Verified Uncertainty Calibration. A. Kumar, P. Liang, T. Ma. NeurIPS 2019.
> > >
> > > [4] Beyond temperature scaling: Obtaining well-calibrated multiclass probabilities with Dirichlet calibration. M. Kull, M. Perello-Nieto, M. Kängsepp, T. S. Filho, H. Song, P. Flach. NeurIPS 2019.
> > >
> > > [5] Multi-Class Uncertainty Calibration via Mutual Information Maximization-based Binning. K. Patel, W. Beluch, B. Yang, M. Pfeiffer, D. Zhang. ICLR 2021.

---

> > > ### Author Response · Authors · 2021-11-29
> > > **Detailed results for the responses**
> > >
> > > (These are detailed results to support the previous post)
> > >
> > > We give results for the 5 datasets where vector scaling works.
> > >
> > > The average ID ECE after vector scaling was 1.37% vs. after temperature scaling was 1.58%. So **vector scaling indeed improves the ECE on in-distribution data**. However, the average OOD ECE after vector scaling was 11.1% vs. after temperature scaling was 9.8%. So **vector scaling did not improve the OOD ECE**.
> > >
> > > In terms of final accuracies, if we use vector scaling the average ID accuracy was 90.8%, and the average OOD accuracy was 77.6% (on the five datasets excluding ImageNet). If we use temperature scaling the average ID accuracy was 91.0%, and the average OOD accuracy was 77.9%. Both methods get the "best of both worlds", and temperature scaling is slightly better, but there **isn't a substantial difference between calibration methods in final accuracy.**
> > >
> > > Full results (will add to our paper):
> > >
> > > ID ECE after vector scaling:
> > >
> > > | datasets     | Ent30     | DomNet    | CIFAR10   | Land      | Crop      |
> > > |--------------|-----------|-----------|-----------|-----------|-----------|
> > > | Cal Standard | 0.9 (0.0) | 2.6 (0.3) | 0.7 (0.0) | 0.9 (0.2) | 1.4 (0.3) |
> > > | Cal Robust   | 0.7 (0.2) | 1.2 (0.2) | 0.8 (0.0) | 1.3 (0.2) | 3.2 (0.2) |
> > >
> > > OOD ECE after vector scaling:
> > >
> > > | datasets     | Ent30      | DomNet     | STL       | Land       | Crop      |
> > > |--------------|------------|------------|-----------|------------|-----------|
> > > | Cal Standard | 19.1 (0.0) | 15.7 (0.3) | 8.5 (0.0) | 17.8 (0.2) | 7.9 (0.3) |
> > > | Cal Robust   | 16.0 (0.2) | 5.4 (0.2)  | 8.4 (0.0) | 7.3 (0.2)  | 5.3 (0.2) |
> > >
> > > ID Accuracies (temp scaling vs vector scaling):
> > >
> > > | datasets      | Ent30      | DomNet      | CIFAR10     | Land        | Crop       |
> > > |---------------|------------|-------------|-------------|-------------|------------|
> > > | Temp Scaled   | 93.7 (0.1) | 91.2 (0.7)  | 97.2 (0.1)  | 77.2 (0.2)  | 95.6 (0.1) |
> > > | Vector Scaled | 93.5 (0.1) | 90.8 (0.2)  | 97.3 (0.1)  | 77.0 (0.4)  | 95.6 (0.1) |
> > >
> > > OOD Accuracies (temp scaling vs vector scaling):
> > >
> > > | datasets      | Ent30       | DomNet      | STL         | Land        | Crop       |
> > > |---------------|-------------|-------------|-------------|-------------|------------|
> > > | Temp Scaled   | 64.7 (0.5)  | 86.1 (0.2)  | 87.3 (0.2)  | 60.8 (0.8)  | 91.3 (0.8) |
> > > | Vector Scaled | 63.7 (0.6)  | 86.1 (0.3)  | 86.8 (0.2)  | 60.4 (1.3)  | 91.0 (0.9) |

---

> > > ### Author Response · Authors · 2021-12-06
> > > **More advanced calibration methods**
> > >
> > > We just wanted to check if our response addressed your concerns about trying out a stronger recalibration method, and about including results for datasets like WILDS? In short, we showed that vector scaling improves ID calibration, but does not improve OOD calibration or accuracy. We also explained that existing robustness methods are not effective in WILDS as reported by the WILDS paper, so we were unable to use the dataset for our research.
> > >
> > > We'd love to hear how we could further improve our work. Thank you for your time!

---

### Official Review · Reviewer_u3S8 · 2021-10-30

**Correctness:** 3
**Technical Novelty And Significance:** 2
**Empirical Novelty And Significance:** 2
**Recommendation:** 5
**Confidence:** 3

**Main Review:**

Strengths
1) A simple algorithm that works in a wide range of datasets.

Weaknesses
1) I'm not convinced why the proposed method should work. I think the authors should provide more explanations why the newly introduced temperature parameter, which is tuned on the validation dataset, should show improved OOD accuracy.
2) I'm not sure about the self-training results in Table 3. compared to the original results in the In-N-Out paper [1]. Maybe releasing the author's implementation can demystify the concerns.
3) The paper lacks any ablation study or analysis to support their claims.

#References
[1] IN-N-OUT: PRE-TRAINING AND SELF-TRAINING
USING AUXILIARY INFORMATION FOR OUT-OF DISTRIBUTION ROBUSTNESS


**Summary Of The Paper:**

UPDATE:

I acknowledge that I've read the author responses as well as the other reviews.

I now understand the empirical merit of the paper. Furthermore, the authors clarified the baselines' performance and add further analysis. Therefore, I raise my score to 5 weak reject. However, I'm still not convinced about the novelty of the paper. In ensemble learning, diversity between ensembles is a major factor of success. Therefore, while it is straightforward that the proposed method outperforms robust+robust and standard+standard, temperature scaling is not a new contribution and I'm still skeptical about how 'surprising' the results are. I hope the authors can clarify further issues about the point.

================================================================================================

The paper studies ensemble methods to achieve improved in-distribution (ID) and out-of-distribution accuracy. The paper proposes a simple fix on concatenating the robust model and the standard model via adopting the idea of classifier-specific temperature parameters, which is optimized on the in-distribution validation dataset. Authors test their methods on various distribution shifts. The proposed method seems to improve the ID-OOD accuracy tradeoff.

**Summary Of The Review:**

While I think the results in the paper are competitive against the baselines, the authors do not give any explanations for why the proposed method should work. Therefore, I'm leaning towards rejecting the paper.

---

> ### Author Response · Authors · 2021-11-12
> **In-N-Out baselines**
>
> Thank you for your useful suggestions---we will respond to the review soon, but wanted to make a quick clarification about the In-N-Out results. The In-N-Out checkpoints and results were taken from the authors' official replication linked in their paper, available at: https://worksheets.codalab.org/worksheets/0x2613c72d4f3f4fbb94e0a32c17ce5fb0 - specifically the row "landcover_in-n-out_iter0" in "Final Results" for each dataset, which they say above the table "corresponds to the In-N-Out algorithm in the paper"

---

> ### Author Response · Authors · 2021-11-21
> **What ablations would help?**
>
> Quick question - what ablations do you think would make the paper stronger? We included a few ablations: 1. Our method without the calibration step, 2. Ensembling only standard or only robust models, 3. Tuning the ensemble weights ID. We're happy to include more ablations based on your suggestions.

---

> ### Author Response · Authors · 2021-11-23
> **Added intuitions, paper has many ablations, added more ablations**
>
> We thank Reviewer u3S8 for taking the time to review the paper.
>
> > The reviewer says a key concern is we "do not give any explanations for why the proposed method should work"
>
> We believe the observation that this method works is already interesting since the method is simpler but competitive with existing tradeoff mitigating methods that use unlabeled data. In general, most observations in deep learning are first reported empirically, and it takes many years to build an understanding. For example, at present we only have a partial understanding of why deep ensembles work, 5-6 years after initial empirical success was reported.
>
> That said, we agree with your point. We have added intuitions and analysis for why the method works in Section 5.3. A key point is that we do not need the models to be calibrated OOD---**a weaker notion of relative calibration** suffices. For example, if the robust model is more accurate and more confident OOD, then we expect that the ensemble will typically use the robust model's predictions OOD. Indeed, Table 8 shows that the relative calibration OOD is much better than the absolute calibration OOD.
>
> Why does calibrating ID improve the relative calibration? Standard deep learning models tend to be overparameterized, and can often get near 0 loss on the training data. As such they tend to be highly overconfident on ID and OOD data. Robustness interventions typically involve additional constraints (projecting out spurious input features, lightweight fine-tuning, extensive data augmentation) and so robust models generally tend to be less overconfident. Calibrating both models, even just ID, makes their confidences more comparable.
>
> This is only a first step towards understanding this phenomena, but we hope this discussion strengthens the paper.
>
> > The reviewer was unsure if we compared fairly with "original results in the In-N-Out paper"
>
> We've clarified that we took their official checkpoints and results. We believe the fact that we achieve competitive results to In-N-Out with a very simple method is a strong selling point of the paper.
>
> > The paper lacks any ablation study or analysis
>
> We believe this review misses our ablations - we ablated every component of our method:
> 1. We ablated different types of ensemble members (standard ensemble, robust ensemble, standard + robust ensemble)
> 2. We ablated the type of ensembling (calibration vs no calibration vs tuning weights ID)
> 3. We've now also added ablations of combining the logits vs softmax probabilities
>
> In addition, we have now also added results for the calibration error and relative calibration of the models.

---

> ### Author Response · Authors · 2021-11-28
> **Responses to follow up questions on surprising aspects of the work**
>
> We thank the reviewer for the follow up questions (in the updated review), and for engaging with us to improve our work.
>
> > Reviewer u3S8 said it's not very surprising that a standard+robust ensemble outperforms a standard+standard ensemble and a robust+robust ensemble, and asked what aspects of our results are surprising or novel.
>
> This is a great point. The comparison with standard+standard or robust+robust ensembles is meant as a sanity check---we agree that this comparison is not surprising.
>
> The three things we found surprising were:
>
> 1\. **Calibrated ensembles eliminate the accuracy tradeoff (instead of just interpolating)** between standard and robust model accuracies.
>
> Recall that ensembles interpolate between the outputs of the standard and robust model. The standard model is worse OOD---so putting a higher weight on the standard model could lead to worse OOD accuracy. Similarly, the robust model is worse ID---so putting a higher weight on the robust model could lead to worse ID accuracy.
>
> Therefore, it appears as if we might interpolate between the standard and robust model's accuracies. What we found surprising was that this tradeoff does not happen---instead we get the "best of both worlds" and surpass the strong ID accuracy of the standard model and OOD accuracy of the robust model.
>
> 2\. **Tuning the ensemble weights on ID data doesn't work well**, but calibrated ensembles do. Some recent works (e.g. [1], [2]) suggest that ID and OOD accuracies can often be correlated, so a natural approach is to select the weight for the standard and robust model that maximizes ID validation accuracy. However, this often does not learn good weights (see Table 4)---the reason is that it assigns too high a weight to the standard model, and performs poorly OOD.
>
> 3\. The other aspect that surprised us is that the method **uses no OOD data and is competitive with more complex methods** like self-training (prior work), which requires a lot of additional unlabeled data to mitigate this tradeoff.
>
> > The reviewer mentioned that temperature scaling is not a new contribution.
>
> We agree that temperature scaling is not new. However, prior work shows that even after temperature scaling on ID data, model confidences are unreliable OOD [3]. Our work highlights that temperature scaling on ID data can still play a useful role for OOD---one intuition is that we only need the models to be relatively calibrated to combine them effectively.
>
> References:
>
> [1] Do CIFAR-10 Classifiers Generalize to CIFAR-10? Benjamin Recht, Rebecca Roelofs, Ludwig Schmidt, Vaishaal Shankar. ICML 2019.
>
> [2] Accuracy on the Line: On the Strong Correlation Between Out-of-Distribution and In-Distribution Generalization. John Miller, Rohan Taori, Aditi Raghunathan, Shiori Sagawa, Pang Wei Koh, Vaishaal Shankar, Percy Liang, Yair Carmon, Ludwig Schmidt. ICML 2021.
>
> [3] Can you trust your model's uncertainty? Evaluating predictive uncertainty under dataset shift. Yaniv Ovadia, Emily Fertig, Jie Ren, Zachary Nado, D Sculley, Sebastian Nowozin, Josh Dillon, Balaji Lakshminarayanan and Jasper Snoek. NeurIPS 2019.

---

> ### Author Response · Authors · 2021-12-06
> **Any other questions**
>
> Dear reviewer - we just wanted to check if we've clarified your questions, and explained what results we found surprising?  We'd love to hear how we could further improve our work. Thank you for your time!

---

### Official Review · Reviewer_VdiR · 2021-11-02

**Correctness:** 3
**Technical Novelty And Significance:** 3
**Empirical Novelty And Significance:** 3
**Recommendation:** 6
**Confidence:** 3

**Main Review:**

The strengths of the submission are that

 — the topic is highly relevant and timely

 — the range of datasets is broad, and the empirical results are fairly promising

Some points that could strengthen the submission are as follows:

 — It would be interesting to investigate qualitatively the OOD cases where the (calibrated) standard model has a higher confidence, as well as a quantitative analysis of how often such events occur. I was hoping to find something along these lines in the Appendix, but it doesn’t look like there’s a supplement. It might also be a good sanity check to look at expected calibration errors for both models.

 — While I believe the range of experiments (datasets + models) is sufficiently broad, methods aimed at distributional robustness such as groupDRO [1] or domain adversarial learning [2] (to name some) might be interesting to study under the proposed method (on whatever datasets they’ve been showcased to have been successful upon).

[1] Distributionally robust neural networks for group shifts, ICLR 2020

[2] Domain generalization with adversarial feature learning, CVPR 2018

**Summary Of The Paper:**

The submission considers a very topical issue given the active ongoing interest in OOD performance — models developed with OOD robustness in mind can come with a significant loss of in-distribution performance, and since we expect to mostly encounter in-distribution settings in deployment (assuming deployment is being done responsibly), we need to find ways to trade-off performances for in-distribution and unexpected situations.

The submission illustrates that a fairly simple approach of Platt-scaling done in-distribution for a “standard” and a “robust” model followed by averaging their predictive distributions can sometimes work pretty well: significant performance improvements appear to be achieved both in- as well as out-of-distribution.

**Summary Of The Review:**

The submission demonstrates that a fairly straightforward calibration-based approach for an ensemble model can lead to good performance in both in- as well as out-of-distribution settings. While the empirical illustrations could certainly be made more comprehensive, and the mechanism of improvements analyzed a bit more closely, I believe this is an interesting enough illustration to justify drawing attention to it.

---

> ### Author Response · Authors · 2021-11-23
> **Added ECE and Relative Calibration numbers**
>
> We thank the reviewer for the positive review and for the helpful suggestions. We believe the simplicity of this method is a strength---especially since it is competitive with prior work like self-training which uses lots of unlabeled data.
>
> The review mentions that the method "can sometimes work pretty well". We wanted to note that the **method works consistently well across the ID and OOD datasets**. On 5/6 ID and 5/6 OOD datasets, calibrated ensembles do better than both the standard and robust models. On the remaining two cases, it closes over 95% of the gap between the standard and robust model, basically performing comparably to the better of the two models.
>
> In the updated paper we have **added the ECE of the standard and robust model**, OOD (Table 6) and ID (Table 7). After calibrating ID, the ID ECE is low, but the OOD ECE is still high. Thank you for the suggestion.
>
> Since we're combining models, the relative confidence of the standard and robust models is also important (e.g. if the standard and robust models are both overconfident by the same amount then ensembling would work well). The newly added Table 8 shows the difference between the average confidence of the standard and robust model OOD, before and after calibration. Ideally, this difference in confidence should match up with the difference in OOD accuracy between the standard and robust model. Calibration improves this difference which provides a partial explanation of its success.

---

> > ### Comment · Reviewer_VdiR · 2021-11-24
> > **Post-rebuttal**
> >
> > Thanks for the responses.
> >
> > The reason I used the word “sometimes” is that there are other large-scale OOD datasets/setups (such as the WILDS benchmarks mentioned by kErF) and robustness methods other than the ones considered in the submission that would provide stronger evidence of the generality of this method. I do believe there is sufficient coverage already though.
> >
> > Thanks for the updates with the calibration analysis. I was more interested in investigating the cases when the ID model is more confident and correct on OOD data and vice-versa. For example, if this happens half the time, it would tell us there is something else about the mechanism of improvements than confidences of the standard/robust models aligning with distribution-shifts (or relative calibrations). Land-Cover seems to suggest alternative mechanisms at play, and averages might be ineffective for going any deeper.
> >
> > Having read the reviews/rebuttals, my rating remains unchanged.

---

> > > ### Author Response · Authors · 2021-11-24
> > > **Thank you**
> > >
> > > Thank you for the quick response, and the helpful suggestions!

---

### Author Response · Authors · 2021-11-23
**Overall Response**

The reviewers agree that we get "strong empirical results provided by the simple yet effective approach" for an "important problem" (REkrF), our work is an "interesting enough illustration to justify drawing attention" (RVdiR), that we give "a simple algorithm that works in a wide range of datasets" (Ru3S8), and "the idea is impactful, easy to implement, and explained clearly" (R7Ez2).

We believe the simplicity is a big strength since the method is simpler but competitive with existing tradeoff mitigating methods that use unlabeled data. We also ablated every component of the method---for example comparing with (1) Tuning the ensemble weights in-distribution, (2) Ensembling standard models or ensembling robust models, (3) Ensembling in the logit versus the probability space. We've added an analysis of the relative calibration of the models, as a first step towards understanding this approach.

The main concerns were:

- Lack of theoretical and conceptual understanding (u3S8 and EkrF): We hope that highlighting this work (which reviewers agree is interesting and effective) spurs efforts into theoretically understanding it better. We believe that most observations in deep learning are first reported empirically, and it takes many years to build an understanding. This is especially true for ensembles, where the community has a limited understanding of why they work even in the traditional IID setting. That said, we have analyzed the "relative calibration" of the models in the newly added Section 5.3, and give intuition for why ensembling and calibration help.

- Additional ablations (e.g. softmax vs logits, R7Ez2): we've added ablations of combining softmax vs logits, and also the ECE and the relative calibration of the models before and after calibration.

- Length of paper: a few reviewers mentioned that we only used 7 of the allotted 9 pages. This was an intentional decision - we felt that the idea was simple and effective, and so we wanted to communicate this more efficiently and save reviewer and reader time. However, we appreciate the feedback and have added more analysis as requested.

---

> ### Author Response · Authors · 2021-11-29
> **Breadth of benchmarks, and using WILDS**
>
> In the follow up discussions, a couple of reviews had a nice suggestion of adding WILDS to to make our experiments more comprehensive. The challenge is that most robustness methods simply don't work on WILDS---the WILDS paper (from ICML 2021) reports that the robustness methods they tried "generally fail to improve over models trained with ERM". Since WILDS is a new dataset, we hope that new robustness methods will emerge that can be used in future work.
>
> We want to emphasize that while our paper is concise, our experiments are broad. We consider 6 datasets, including popular domain adaptation benchmarks (DomainNet, CIFAR $\to$ STL), robustness benchmarks (BREEDS, ImageNet $\to$ ImageNet-R), and two real world satellite remote sensing datasets including a time series dataset, which were used by prior work on mitigating tradeoffs. We consider three robustness interventions, and multiple architectures (vision transformers, ResNet-50, time series convolutions).

---

### Decision · Program_Chairs · 2022-01-20

**Decision:**

Reject

**Comment:**

This is a borderline case and it's quite difficult to decide the recommendation. The paper works on a critically important problem, namely removing or reducing the in-distribution accuracy drop when we need to also take the out-of-distribution accuracy into account. The proposed method is simple and it works, which is great. However, as the reviewers discussed, the demonstrated applications are not very representative, and the authors should consider more popular setups of few-short learning and even other forms of domain generalization. Furthermore, adversarial examples are also OOD (in most cases, since the ID manifolds are thin films and the attacks can easily go out of the ID manifolds), it would be great if adversarial accuracy can be incorporated as a case of OOD accuracy. Since there is still room for improvement, we hope the paper would benefit from a cycle of revisions for a re-submission.